# A chicken DNA methylation clock for the prediction of broiler health

Günter Raddatz[1], Ryan J. Arsenault[2], Bridget Aylward [2], Rose Whelan [3], Florian Böhl[4] & Frank Lyko [1✉]

The domestic chicken (*Gallus gallus domesticus*) is the globally most important source of commercially produced meat. While genetic approaches have played an important role in the development of chicken stocks, little is known about chicken epigenetics. We have systematically analyzed the chicken DNA methylation machinery and DNA methylation landscape. While overall DNA methylation distribution was similar to mammals, sperm DNA appeared hypomethylated, which correlates with the absence of the DNMT3L cofactor in the chicken genome. Additional analysis revealed the presence of low-methylated regions, which are conserved gene regulatory elements that show tissue-specific methylation patterns. We also used whole-genome bisulfite sequencing to generate 56 single-base resolution methylomes from various tissues and developmental time points to establish an LMR-based DNA methylation clock for broiler chicken age prediction. This clock was used to demonstrate epigenetic age acceleration in animals with experimentally induced inflammation. Our study provides detailed insights into the chicken methylome and suggests a novel application of the DNA methylation clock as a marker for livestock health.

[1] Division of Epigenetics, DKFZ-ZMBH Alliance, German Cancer Research Center, Im Neuenheimer Feld 580, 69120 Heidelberg, Germany. [2] Department of Animal and Food Sciences, University of Delaware, Newark, DE 19716, USA. [3] Animal Nutrition Services, Nutrition & Care, Evonik Operations GmbH, Fort Dunlop, Fort Parkway, B24 9FE Birmingham, Great Britain. [4] Creavis, Evonik Operations GmbH, Rodenbacher Chaussee 4, 63457 Hanau-Wolfgang, Germany. ✉email: f.lyko@dkfz.de

The domestic chicken (*Gallus gallus*) is the globally most ubiquitous animal livestock and a significant source of commercially produced meat and eggs[1]. Factors that influence the growth, pathogen resistance and meat quality of chickens are of considerable scientific and economical interest[2]. Extensive analyses have been performed to establish the underlying genetic framework and have resulted in a major increase in livestock improvement[3,4]. However, novel approaches are required to meet the expected 70% increase in the demand of meat by 2050[5]. In this context, epigenetic modification patterns[6] represent a novel and potentially resourceful complement to the prevailing genetic frameworks.

Epigenetic modification patterns allow the analysis of context-dependent information on DNA[7]. DNA methylation represents the best-understood epigenetic modification and is characterized by high specificity for CpG dinucleotides in animal genomes[8–10]. DNA methylation is catalyzed by the DNMT family of enzymes, which has diverse functions in epigenetic gene regulation[11]. Conserved homologues include the canonical DNA methyltransferases DNMT1 and DNMT3, the tRNA methyltransferase DNMT2 and the catalytically inactive DNMT3L cofactor, which is required for effective DNA methylation of repetitive elements in the mammalian germline[12,13]. However, comparably little is known about the genome-wide DNA methylation patterns of non-mammalian vertebrates.

Based on its substantial capacity for biomarker development and stock improvement, several recent studies have started to elucidate the chicken methylome. While initial approaches were mostly based on indirect methods[14–17], more recent studies also used whole-genome bisulfite sequencing to establish chicken methylomes at single-base resolution[18–21]. However, some initial results, such as the presence of non-CpG methylation and the relative hypomethylation of repeats[18] appeared contradictory to other known vertebrate methylomes. Similarly, the dynamics of DNA methylation in chicken are only beginning to be investigated[21] and comprehensive datasets for the development of epigenetic biomarkers are presently not available.

A particularly prominent example for an epigenetic biomarker is provided by DNA methylation clocks[22,23], which are sets of CpG-specific methylation values that, in combination with a mathematical algorithm, estimate the age of the DNA source[24]. While the methylation level of individual CpGs usually exhibits a relatively weak correlation with age, methylation clocks are composite multivariate biomarkers that have been shown to accurately measure age in humans and mice[24]. DNA methylation clocks have also been constructed for a number of additional animals (e.g., chimpanzees, dogs, wolves, whales), but remain to be fully developed and understood[25]. For example, animal clocks are often based on single tissues and on limited numbers of methylation marks. Furthermore, the functionality of animal DNA methylation clock and their application potential remains poorly understood.

Interestingly, methylation clocks have also revealed many examples where DNA methylation age and chronological age are divergent. These differences have been interpreted to reflect different speeds of biological aging, and can even be used to predict all-cause mortality[26]. Furthermore, many pathological conditions are associated with epigenetic age acceleration in humans, while anti-aging diets in mice have been shown to result in age deceleration[24,27]. The available findings in humans and mice thus suggest that differences between methylation age and chronological age can be used as health biomarkers, with age acceleration indicating poor health and age deceleration indicating good health. Health biomarkers are also emerging tools for agriculture, as they facilitate the monitoring of large groups of animals and provide objective quality assurance.

The broiler chicken presents an important challenge for DNA methylation clock development, as it combines considerable economic importance with a short, commercially determined lifespan of only 42 days. Furthermore, no DNA methylation clocks have been developed so far for any representative bird species. We have now analyzed genome-wide DNA methylation patterns of broiler chickens across various tissues and developmental timepoints. Our results show a dynamic methylome that was used for the development of a multi-tissue DNA methylation clock and applied for broiler health analysis.

## Results

**The DNA methylation machinery of the chicken**. A detailed BLAST analysis detected three chicken homologs for canonical DNMT enzymes (Fig. 1a): DNMT1 (Q92072), DNMT3A (Q4W5Z4) and DNMT3B (Q4W5Z3). In addition, we detected a conserved homologue of the DNMT2 tRNA methyltransferase (Q4W5Z2). However, a BLAST search of the 24 DNMT3L sequences contained in the Swiss-Prot database (including an alligator homolog) against the current chicken genome assembly revealed no significant homology. This suggests that DNMT3L is not conserved in chicken. Indeed, a subsequent analysis of 421 automatically annotated genomes from NCBI/refseq showed that DNMT3L is conserved in most mammals and in many reptiles, with the exception of birds and monotremes, amphibians and fish (Fig. 1b). This finding establishes chicken as an archetypal species for a DNMT3L-deficient vertebrate methylome.

**Characterization of the chicken methylome**. In an initial analysis of published whole-genome bisulfite sequencing datasets (Table S1), we compared the chicken methylation landscape to the mouse (mammal), elephant shark (non-mammalian vertebrate) and sea squirt (invertebrate at the base of vertebrate evolution). The results showed that the overall methylation patterning in chicken is similar to the mouse (DNMT3L present) and elephant shark (DNMT3L absent), but different from the invertebrate sea squirt (Fig. 1c). More specifically, the chicken methylome was characterized by dense methylation, similar to the mouse and elephant shark methylomes[28], while the sea squirt methylome was characterized by patches of moderate methylation[29]. These findings indicate that the absence of DNMT3L does not have a major impact on the methylation ratio distribution of the somatic methylome. In addition, our results also indicate that the chicken shares the general methylation pattern of other vertebrates, including mammals.

In mammals, DNA methylation patterns are highly tissue-specific[30]. This tissue-specificity is widely regarded to reflect a key function of DNA methylation, i.e., the establishment and maintenance of cell-type-specific epigenetic programs[6]. Surprisingly, however, very little is known about the evolutionary conservation of tissue-specific DNA methylation and it has been shown that gene methylation patterns in tunicates and crayfish are largely tissue-invariant[29,31]. To elucidate tissue-specific DNA methylation in chicken, we integrated additional published whole-genome bisulfite sequencing (WGBS) datasets (Table S1) into our analysis. Initial data analysis showed a relatively high level of methylation and substantial specificity for CpG dinucleotides. The analysis of genomic features showed relatively high methylation levels at exons, introns and 3'-UTRs (Fig. 2a). We also found repeats to be highly methylated (Fig. 2b), which resolves ambiguous findings from previous analyses[18]. Gene promoters and 5'-UTRs appeared lowly methylated, consistent with methylation patterns observed in other vertebrates[6] (Fig. 2a, c). Of note, we observed moderate, but highly significant ($P = 2.2 \times 10^{-16}$, Wilcoxon rank sum test) differences in the methylation levels of genomic features between lung, breast muscle and sperm samples (Fig. 2a). Most notably, the sperm methylome appeared less methylated compared to somatic

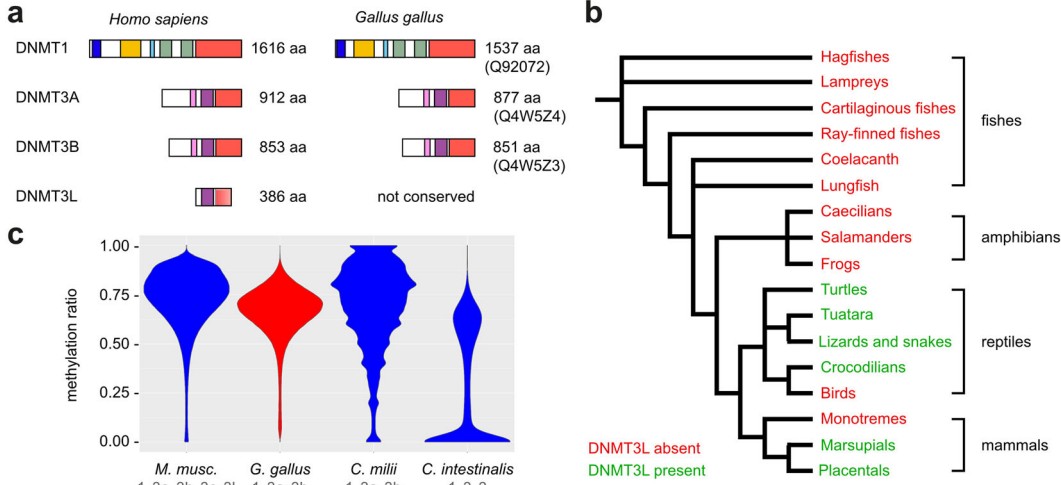

**Fig. 1 A conserved DNA methylation system in chicken. a** Conservation of canonical DNMT enzymes in chicken. Conserved domains of animal DNMTs are shown in different colors. All DNMTs: catalytic domain (red). DNMT1: DMAP1 binding domain (dark blue), replication foci targeting domain (orange), CXXC domain (turquoise), BAH domain (green). DNMT3: PWWP domain (pink), ADD domain (purple). DNMT3L is a catalytically inactive DNMT3 variant that lacks the N-terminal part of the regulatory domain (including the PWWP domain) and the C-terminal part of the catalytic domain. **b** Evolutionary conservation of DNMT3L in vertebrates. The phylogenetic tree is based on 421 automatically annotated genomes from NCBI/refseq. **c** Interspecific comparison of somatic methylation landscapes. Violin plots show methylation ratios of 2 kb sliding windows covering the entire genome. The uneven plot shape for the elephant shark (*Callorhinchus milii*) is related to the relatively low sequencing coverage. The DNMT homologs of individual organisms are indicated under the respective species names.

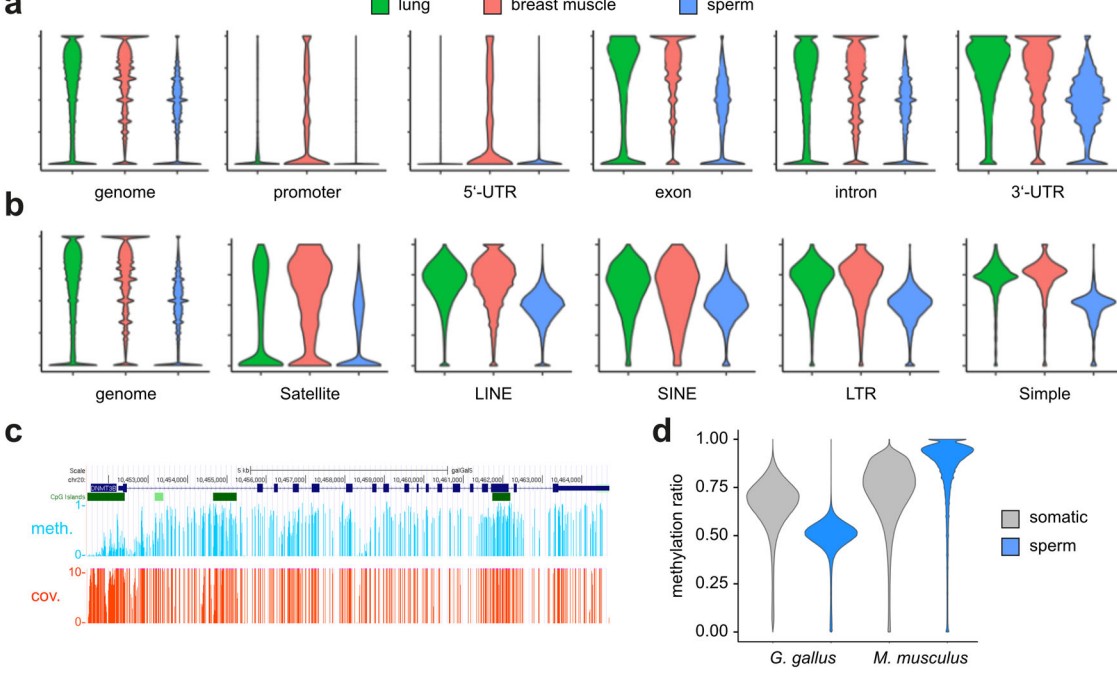

**Fig. 2 Methylation analysis of genomic features. a** Violin plots showing methylation levels of different genomic features in three different tissues. All differences between tissues are statistically significant ($P = 2.2 \times 10^{-16}$, Wilcoxon rank sum test), except the difference between lung and breast muscle at 3'-UTRs. **b** Violin plots showing methylation levels of different repeat classes in three different tissues. **c** Representative gene body methylation pattern. The methylation track shows DNA methylation levels (light blue) and sequencing coverage levels (red) for all CpGs of the chicken DNMT3B locus (chromosome 20) in lung. Sequencing coverages were cut off at >10. **d** Violin plots showing methylation ratios of 2 kb sliding windows. Representative somatic datasets are from chicken lung and mouse intestine.

methylomes (Fig. 2d). This contrasts the sperm DNA hyper-methylation in mammals[32] (Fig. 2d) and may be related to the absence of DNMT3L in chicken (see discussion). Taken together, these findings provide a first indication that DNA methylation in chicken is tissue-specific.

**Dynamic methylation of the chicken genome.** Low-methylated regions (LMRs) represent a key feature of the tissue-specific mammalian methylome[33]. As LMRs reflect the binding of transcription factors, they can change dynamically during cell fate specification and thus contain important information about

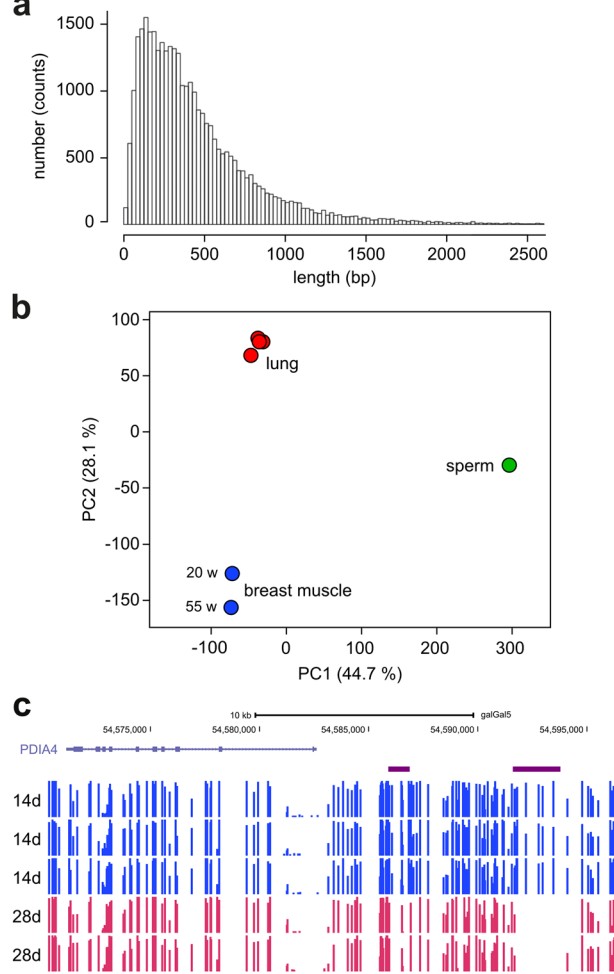

**Fig. 3 Identification of low-methylated regions (LMRs) in the chicken methylome. a** Size distribution of chicken LMRs. **b** Principal component analysis of LMR methylation patterns from different tissues, strains, and age groups. **c** Age-related differential methylation of LMRs in the jejunum of broiler chickens. Representative methylation tracks for the *PDIA4* gene in d14 (blue) and d28 (crimson) samples. The position of LMRs is indicated by purple horizontal bars. Note the pronounced differential methylation between d14 and d28 for the rightmost LMR.

epigenetic programming. Because many features of the mammalian methylome appeared to be conserved in the chicken methylome, we used an LMR identification tool (MethylSeekR) on the published chicken datasets (Table S1). This identified 47,000 LMRs covering 17 Mbp of the chicken genome (Fig. 3a). Typical LMRs extended over several hundred bp and were invariably associated with transcription factor binding sites (Fig. S1). Consistently, many conserved transcription factor binding sites showed a highly significant enrichment within LMRs (Table S2). These findings strongly suggest the conservation of LMRs in the chicken methylome. As LMR methylation patterns can be considered a reliable proxy of epigenetic programs, we subsequently investigated their ability to separate methylomes from different tissues. Indeed, principal component analysis of LMR methylation patterns achieved robust separation between the three tissues analyzed (Fig. 3b).

To further explore the dynamics of DNA methylation in chicken, we generated replicate WGBS datasets from the jejunum of broilers from two different age groups (d14 and d28, Table 1). This provided chicken methylation maps with an unprecedented resolution power for further analysis. Subsequent analysis

identified 33,422 LMRs in these datasets. Pairwise comparisons of LMR methylation ratios between the d14 and d28 groups revealed 1728 differentially (ratio difference >0.1) methylated LMRs (Fig. 3c), of which 964 were hypermethylated in d28 intestines, while 764 were hypomethylated in d28 intestines. These findings suggest that DNA methylation patterns in chicken can show dynamic age-dependent changes.

**A chicken DNA methylation clock**. DNA methylation clocks represent powerful biomarkers for age estimation[24]. To establish a multi-tissue DNA methylation clock for the relatively short lifespan of the broiler chicken (0–42 days), we performed additional WGBS on 36 samples from four tissues (breast muscle, ileum, jejunum, spleen), with ages ranging between 3 and 35 days. The tissues were chosen because of their economic importance (breast muscle), and because of their importance for intestinal health (ileum, jejunum) and overall health (spleen). The upper timepoint (35 days) was determined by regulatory standards for curative treatment (up to 1 week before slaughtering). Overall, our approach is comparable to other known animal DNA methylation clocks, but shows superior CpG coverage, as it is based on whole-genome bisulfite sequencing, rather than the more limited reduced representation bisulfite sequencing or PCR assays used for other studies (Table 1, Table S3).

PCA analysis of LMR methylation patterns (Fig. 4a) revealed different offsets for different tissues, possibly reflecting different stages of tissue maturation for individual tissues. We therefore implemented a normalization step to correct for this feature (see Methods for details). In addition, we also removed all CpGs that were associated with sex chromosomes or with single-nucleotide polymorphisms (Fig. 4b). We then used a penalized regression model to regress the chronological age on 67,651 LMRs in our training set of 36 samples. The alpha value was varied in a range between 0 and 1 and chosen as 0.9 (elastic net regression), because this value led to a fit that was close to the best fit and a manageable amount of LMRs (Fig. S2). This identified a set of 32 LMRs together with corresponding beta values, which define the weights for these LMRs used in the chicken methylation clock (Table S4). For example, a highly weighted clock LMR was found in the chicken *Igfbp3* gene, which is involved in insulin signaling. This LMR showed a pronounced age-dependent increase in DNA methylation (Fig. 4c). Six-fold cross-validation of the LMR clock showed a root mean square error of 1.6 days, suggesting appreciable accuracy.

In parallel analyses, we also used a penalized regression model to regress the chronological age on 257,913 highly covered CpGs (see Methods for details) in our training set of 36 samples. An alpha value of 0.7 identified a set of 45 CpGs (Tab. S5) that showed a root mean square error of 3.4 days after 6-fold cross-validation. A comparison of LMR and CpG clocks showed similar distribution patterns of the respective clock markers in promoters, gene bodies and intergenic regions. Interestingly, both clocks showed a strong feature enrichment in promoters (Fig. 4d), which is consistent with the known epigenetic regulatory role of promoter DNA methylation[9]. Furthermore, we also validated our clocks by generating whole-genome bisulfite sequencing datasets of 6 samples (breast muscle) in two age groups (14 and 28 days) from a completely independent animal trial. The results showed accurate age prediction (Fig. 4e) with root mean square errors of 2.6 and 3.4 days, respectively, for the LMR clock, and 2.3 and 3.7 days, respectively, for the CpG clock. While these findings suggest overall similar performances for the LMR and CpG clocks, the LMR clock is based on coordinated methylation changes over several neighboring CpGs, which likely explains its higher prediction accuracy in the cross-validation.

**Table 1 Whole-genome bisulfite sequencing datasets generated in this study.**

| Tissue | Age (days) | No. of samples | Coverage | Conversion | Used for |
|---|---|---|---|---|---|
| Jejunum | 14 | 3 | 122.0× | >99.9% | LMR mapping |
| | 28 | 2 | 65.8× | >99.9% | LMR mapping |
| Ileum | 3 | 3 | 54.5× | >99.9% | Clock training |
| | 15 | 3 | 37.0× | >99.9% | Clock training |
| | 34 | 3 | 49.0× | >99.9% | Clock training |
| Breast muscle | 3 | 3 | 52.7× | >99.9% | Clock training |
| | 15 | 3 | 37.6× | >99.9% | Clock training |
| | 34 | 3 | 34.6× | >99.9% | Clock training |
| Spleen | 3 | 3 | 54.6× | >99.9% | Clock training |
| | 15 | 3 | 38.9× | >99.9% | Clock training |
| | 34 | 3 | 34.9× | >99.9% | Clock training |
| Jejunum | 14 | 3 | 55.4× | >99.9% | Clock training |
| | 16 | 3 | 54.7× | >99.9% | Clock training |
| | 34 | 3 | 57.7× | >99.9% | Clock training |
| Breast muscle | 14 | 3 | 55.9× | 98.2% | Clock validation |
| | 28 | 3 | 56.7× | >99.5% | Clock validation |
| Jejunum | 14 | 3 | 43.3× | >99.9% | Clock performance |
| | 16 | 3 | 47.4× | >99.9% | Clock performance |
| | 35 | 3 | 46.7× | >99.9% | Clock performance |

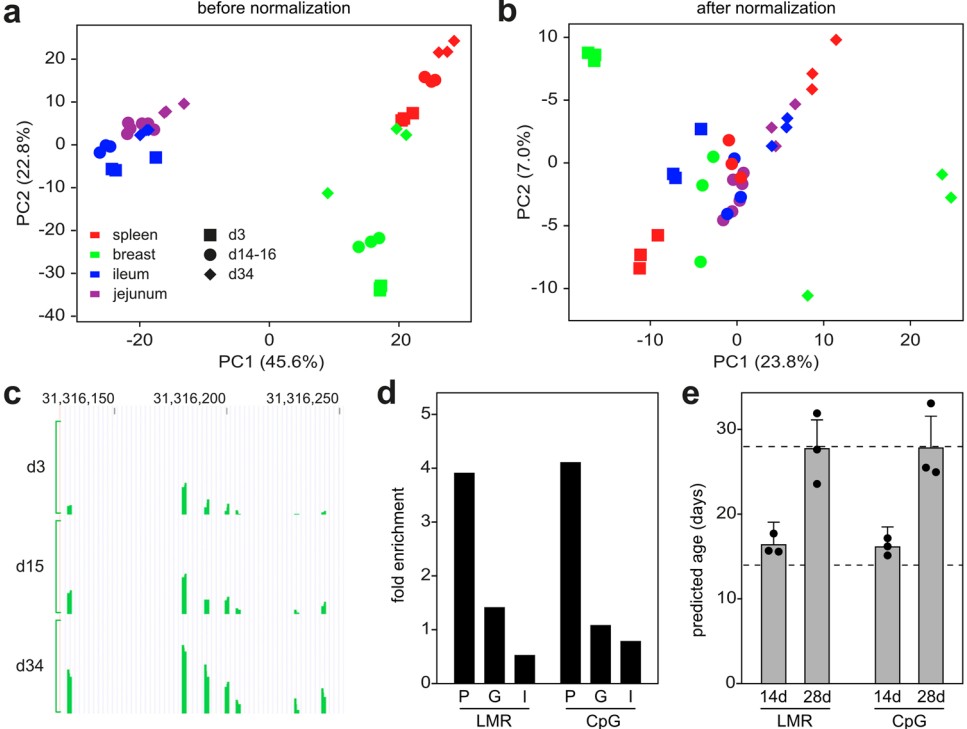

**Fig. 4 Establishment and validation of a multi-tissue DNA methylation clock for broiler chicken. a** PCA analysis based on the DNA methylation patterns of 67,651 LMRs in the training set ($N = 36$ samples). **b** PCA based on LMR methylation patterns after filtering and normalization. **c** Example of a highly weighted clock LMR from the *Igfbp3* gene body (#5 in Tab. S4), showing progressive age-related methylation. Coordinates refer to chromosome 2 of chicken genome assembly (version 5.0). Green lines indicate methylation levels from 0 to 1. **d** Enrichment of clock LMRs and clock CpGs in specific genomic features. P promoters, G gene bodies, I intergenic regions. **e** Validation of LMR-based and CpG-based age prediction in breast muscle tissue from 6 independent animals belonging to 2 age groups (14 days and 28 days). Error bars indicate RMSEs.

**The DNA methylation clock predicts broiler health.** Finally, we used both clocks to test their ability to measure broiler health. It has been shown that inflammation can result in accelerated methylation aging in humans[34] and intestinal inflammation is a well-known health issue for broiler chicken[35–37]. We therefore analyzed intestinal (jejunum) samples from a trial where systemic inflammation was induced with multiple doses of a CpG

oligonucleotide (Fig. 5a), which represents an established and potent protocol for the activation of cell-mediated immune responses[38]. To monitor the immunological status of the animals, we analyzed jejunal tissue via a species-specific kinome peptide array (see Methods for details). Kinome arrays are designed to measure the phosphorylation-based signaling of a broad range of cellular responses. Compared to individual biomarkers, like the

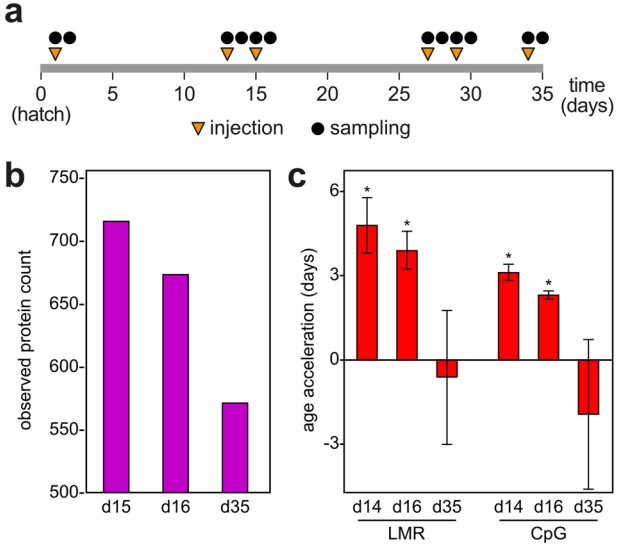

**Fig. 5 Intestinal inflammation is associated with age acceleration in broiler chickens. a** Schematic outline of the inflammation trial. After hatching (day 0), animals were periodically injected with CpG (or control GpC) oligonucleotide, as shown (orange tringles). At several timepoints (black circles), animals were euthanized and tissue was collected for analysis. **b** Inflammatory kinase activation in jejunum samples with experimentally induced inflammation. The number of observed differentially phosphorylated proteins in the top 10 immune related GO terms were counted for CpG injected broiler chickens at each day. At day 35 post-hatch, a substantial reduction in protein phosphorylation changes related to immunity was observed. **c** Age acceleration in jejunum samples with experimentally induced inflammation. The specific age acceleration in the d13-16 age group matches the immunological phenotype. Asterisks indicate statistically significant differences in relation to day 35 ($P < 0.05$, t-test).

expression of pro-inflammatory cytokines or the presence of acute phase proteins, they allow a more comprehensive and more sensitive analysis across many species[39]. Gene Ontology analysis of the significant proteins represented by the peptide targets was used to characterize the immune response (Table S6) and revealed that the most pronounced immune response occurred at day 15, with a slight reduction at day 16 and a strong reduction at day 35 (Fig. 5b). Notably, these relative immune responses matched the results obtained with the DNA methylation clock from same sample groups, as we observed a specific and statistically significant age acceleration at days 14 and 16, both for the LMR clock and for the CpG clock (Fig. 5c). These results illustrate how the DNA methylation clock can be used to analyze the health status of the broiler chicken.

## Discussion

The use of molecular markers plays an important role for the improvement of livestock performance. However, this field has been largely restricted to genetic markers that facilitate selective breeding. Our results establish key features of the chicken epigenome and its capacity for dynamic, tissue-specific modulation. We provide detailed insight into the chicken methylome, based on high-quality datasets from various tissues and developmental timepoints. Furthermore, we establish and validate a DNA methylation clock for broiler chickens that can be used to measure their health status.

Our study clarifies the DNA methylation landscape of the chicken and sheds light on specific features, such as the lack of a DNMT3L gene and the global hypomethylation of sperm DNA.

DNMT3L lacks the complete set of catalytic motifs found in canonical DNA methyltransferases[12]. Indeed, the enzyme is catalytically inactive, but can act as an important cofactor for DNA methylation in mammals[40]. Our results show that DNMT3L is absent from bird and monotreme genomes, but also from fish and amphibian genomes. This suggests that the gene was gained by gene duplication in the common amniote ancestor and then lost during the evolution of the bird and monotreme lineage. DNMT3L is a key factor for the comparably high methylation of mammalian sperm DNA[13]. This contrasts the relatively low methylation levels observed in chicken sperm, and the non-conservation of DNMT3L in the chicken genome provides a sensible mechanistic explanation.

Transcription factors play an important role in the regulation of cell-type-specific gene expression programs. Furthermore, the binding of many mammalian transcription factors has been shown to be modulated by DNA methylation[41]. LMRs correspond closely to transcription factor binding sites in mammals[33] and our results show that this feature is conserved in chicken. Furthermore, we found that chicken LMR methylation patterns are highly tissue-specific, which is again consistent with the mouse. LMRs thus represent a relatively small subfraction of the genome that is considerably enriched for dynamically methylated sites. This feature can be used for the resolution of distinct epigenetic states.

Chicken LMRs also proved useful for the development of a chicken DNA methylation clock that can predict the age of individual samples with high accuracy. Of note, accurate age prediction was achieved in a very short timeframe, the commercially restricted broiler chicken lifespan of 42 days. A key step towards achieving this accuracy was the normalization of methylation values, which is necessitated by the different aging trajectories of individual tissues. This tissue-specificity of methylation clocks was also noted in the original publication describing the human DNA methylation clock[23]. Differences in tissue maturation can be pronounced in newborn and developing organisms[42] and thus particularly relevant for the broiler chicken, where development is limited to the juvenile stage. We have developed a solution of this problem by normalization of methylation values to enable the comparison of different tissues. The training of clocks for individual tissues and subsequent normalized combination of the prediction results is likely to further improve clock performance, but requires the availability of larger datasets.

We also compared the performance of our LMR clock to a clock that is based on the selection of individual CpGs. This CpG clock showed a lower prediction accuracy compared to the LMR clock (3.4 days vs. 1.6 days), which is likely to reflect the higher noise susceptibility of isolated CpGs. Also, an LMR clock allows a better conceptual interpretation of its markers, as LMRs represent transcription factor binding sites[33]. This facilitates our biological understanding of DNA methylation clocks.

Finally, the need for poultry health monitoring, especially for the intestine, is well recognized and there is an ongoing discussion about appropriate biomarkers[43]. Our approach combines different advantages for poultry health monitoring, as it can monitor various diseases that affect the speed of the DNA methylation clock. It is also robust against age-related influences, which is important in a dynamically developing tissue like the intestine. As our clock has been trained on multiple tissues, it is likely that the observed inflammation related age acceleration can also be detected in other tissues. Future work should analyze the effect of additional poultry diseases on the chicken DNA methylation clock and also explore the suitability of DNA methylation clocks for the health monitoring of other agricultural livestock.

In conclusion, our study establishes the chicken as a key example for DNMT3L-deficient vertebrate methylomes. Furthermore, our results provide detailed insight into its dynamic regulation at transcription factor binding sites. This information was used to construct a chicken DNA methylation clock, which allows accurate age prediction in broiler chickens. Finally, we show epigenetic age acceleration for animals where inflammation was induced experimentally and thus establish DNA methylation age as a marker for their health status.

## Methods

**Animals and tissue samples**. *Gallus gallus domesticus* (Ross308) was used for all experiments. Animal experiments were approved by the University of Delaware Institutional Animal Care and Use Committee #86R-2019-2. Sampled tissues and sampling timepoints are specified in Table 1.

**Phylogenetic analysis of DNMT3L**. Annotations of 421 genomes automatically annotated with the NCBI eukaryotic gene prediction tool GNOMON were downloaded from the NCBI. Genomes having an annotated DNMT3L gene were extracted from this set and assigned to their phylogenetic position in a standard phylogenetic tree.

**Whole-genome bisulfite sequencing**. DNA was prepared using the Invitrogen PureLink genomic DNA Isolation Minikit, following the manufacturer's instructions. Samples were stored in nuclease-free, DEPC treated water with 10% (v/v) TE buffer. Whole-genome bisulfite sequencing was performed by the DKFZ Genomics and Proteomics Core Facility using standard protocols.

**Read mapping**. Reads were trimmed and mapped with BSMAP[44] version 2.5 using the assembly version 5.0 of the chicken (*Gallus gallus*) genome as reference sequence. Duplicate reads were removed using the Picard tool. Methylation ratios were determined using a Python script (methratio.py) distributed with the BSMAP package. For all further analysis, only CpGs covered by at least three reads were considered.

**Methylation data analysis**. Violin plots were created using the command geom_violin() of the R-package ggplots2. As input, averaged methylation ratios over a 2 kb sliding window were used. Annotation of the genes contained in the galGal5 genome was taken from the Ensembl annotation. Promoters were defined as 1000 bp upstream of the transcription start site. Definitions of repeats contained in the galGal5 genome, including class and family of repeats, were taken from the repeatmasker-based track provided by the UCSC genome server.

**LMR analysis**. Low-methylated regions (LMRs) were defined using the tool MethylSeekR[45] individually for the different tissues (lung, breast muscle, sperm). The resulting sets of LMRs were pooled. This identified a total of 47,012 LMRs covering 17 Mbp of the chicken genome.

**Transcription factor binding site analysis**. The set of 47,012 LMRs was subjected to a transcription factor binding site analysis using the tool Homer[46] and the set of known vertebrate transcription factor binding matrices.

**Filtering and normalization of the methylation data as input for the clock**. We removed all CpGs that were associated with sex chromosomes. We also filtered out all CpGs that are listed as SNPs for the *Gallus gallus* genome in the dbSNP database (https://www.ncbi.nlm.nih.gov/snp/). For the CpG clock, we restricted the analysis to CpGs that showed a strand specific coverage of greater than 10 in every of the sequenced samples, resulting in a set of 257,913 CpGs. For the LMR clock, we restricted the analysis to CpGs within low-methylated regions that showed a strand specific coverage of greater than 5 in every of the sequenced samples, resulting in a set of 67,651 LMRs.

The average methylation values of these LMRs were computed and normalized by computing (for every LMR) the average value over all samples from the same tissue and subtracting this value from the value of this LMR (in case of the CpG clock by computing for every CpG the average value over all samples from the same tissue and subtracting this value from the value of this CpG). The rationale for this approach is illustrated in Fig. 4a, showing the first two principal components of a PCA of the LMR methylation data. PC2 shows a strong positive correlation with the age of the subjects ($r = 0.466$) whereas PC1 does not show any correlation with the age of the subjects ($r = -0.005$). This was interpreted to reflect that PC2 represents the age of the subjects, with a higher age corresponding to a higher value of the sample on PC2. However, even the oldest samples of breast muscle tissue still showed a smaller value than the youngest samples of spleen tissue, although the order within the set of samples of a specific tissue is largely correct. This indicates a tissue-specific "offset" for the positioning in the age-reflecting PC2, which is probably caused by different maturation stages for different tissues at certain timepoints in the early life phase of the chickens. As this offset is likely to affect the training of the methylation clock algorithm, a corresponding correction was introduced.

**Establishment of chicken DNA methylation clocks**. To establish a chicken methylation clock a penalized regression model (implemented in the R-package glmnet[47]) was applied to regress the chronological age of the animals on the normalized methylation values of the CpG probes. This approach, which computationally assigns weights to the set of CpG probes and thus selects an optimized set of markers, was established in the seminal paper by Horvath[23] and has since been applied in nearly all studies on DNA methylation clocks. In the case of the LMR clock, a penalized regression model was applied to regress the chronological age of the animals on the normalized average methylation values of the LMRs.

**Clock characterization**. The enrichment of clock markers in specific genomic features was calculated by dividing the fraction of markers located in the specific feature by the fraction of this feature in the complete chicken genome sequence. Promoters were defined as regions 1 kb upstream of the transcription start site.

**Inflammation trial**. One hundred eighty animals were assigned to one of two groups, CpG treatment or GpC control. The animals were given their first i.p. injection of either CpG or the control GpC as a 25-μg dose in 0.2 ml of 0.01 M sterile phosphate buffered saline (PBS) 1 day after hatching. A detailed injection and sampling outline is shown in Fig. 5a. Five animals from each group were sacrificed at each collection time point via cervical dislocation and tissue samples were preserved in RNAlater, stored at 4 °C overnight, then moved to −20 °C until used for analysis.

**Kinome peptide array**. The peptide array protocol carried out as previously described[48]. More specifically, 40 mg tissue samples were homogenized by a Bead Ruptor 24 homogenizer in 100 μl of lysis buffer containing protease inhibitors. Homogenized samples were then mixed with an activation mix containing ATP and applied to the peptide arrays. Arrays were incubated in a humidity chamber at 37 °C with 5% $CO_2$ allowing kinases to phosphorylate their target sites. Samples were then washed off the arrays and a florescent phosphostain was applied. Stain not bound to phosphorylated sites was removed by a destaining process. Arrays were then imaged using a Tecan PowerScanner microarray scanner at 532–560 nm with a 580-nm filter to detect dye fluorescence. Array images were gridded using GenePix Pro software (Molecular Devices, San Jose, CA) and the spot intensity signal was collected. Florescent intensities for treatments were then compared with controls using the Platform for Intelligent, Integrated Kinome Analysis (PIIKA 2)[49]. The resulting data output was then used in the Search Tool for the Retrieval of Interacting Genes/ Proteins (STRING)[50] database to pinpoint changes in protein-protein interactions and signal transduction pathways.

## Data availability
All sequencing data are available from the GEO database under accession number GSE146620.

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

## Acknowledgements

We thank the DKFZ Genomics and Proteomics Core Facility for sequencing services and Walter Pfefferle for his continuous support.

## Author contributions

G.R. acquired, analyzed and interpreted data, R.A. conceived the animal trials and analyzed and interpreted data, B.A. acquired data, R.W. conceived the animal trials, F.B. conceived the study, F.L. conceived the study and wrote the manuscript with contributions from the co-authors.

## Funding

## Competing interests

The authors declare the following competing interests: R.W. and F.B. are employees of Evonik AG. F.L. received consultation fees from Evonik AG. G.R., R.W., F.B., and F.L. are currently applying for patents related to the contents of this manuscript.
