## [Peer Review File · Communications Biology]

Reviewers' comments:

Reviewer #1 (Remarks to the Author):

Authors produced a huge amount of whole-genome bisulfite sequencing data to DNA methylation markers for age prediction and health monitoring of broiler chicken. They submit genome conservation of DNMT3L to show reduced DNA methylation level in *G. gallus* sperm. In order to select methylation markers, low-methylated regions (LMRs) was used for age prediction and health monitoring. Accuracy still needs to be improved by high-depth whole-genome bisulfite sequencing, larger samples, and better methods of prediction such as deep learning.

The purpose and results of this study is clear and easy to follow research design and its outcomes.

1. Because of small numbers of tissues and individuals, they selected only tens of methylation sites, which are relatively small considering human CpGs (<https://www.nature.com/articles/s41576-018-0004-3/>). Only jejunum was used for LMR mapping, and essentially two groups (ileum+jejunum vs. spleen+breast) for clock mapping. This study successfully found clock markers but in this current status, it would be only candidate for age prediction and health monitoring. If one choose small numbers of markers, a few markers can have large effects (LMR marker chr2:91174539-91175128, chr6:8416238-8416588, chr13:13146982-13147888). Also, some of them showed very little weights. Authors need to thoroughly investigate effects of individual markers, statistical significances and biological meanings. We can also see some high effect markers in clock CpGs.

2. Relationship between inflammation and methylation is well known, they also tried to validate LMR clock for experimentally introduced inflammation. Because their model is only focused in jejunum sample for LMR clock, they successfully showed as shown in figure 4e,f. However, it is not clear whether those putative markers also work in another tissue.

3. They only used penalized regression models for their study. First, selection of markers are not clearly defined. In machine learning technique, one can analyze marker effect and find optimized markers. If there are interactions (nonlinear relationships) between markers, instead of simple regression model, one can apply advanced models, i.e. deep learning methods.

Reviewer #2 (Remarks to the Author):

A chicken DNA methylation clock for livestock health monitoring

Comments for the Author:

This paper is very interesting and useful. Firstly, confirming that chicken is a key example for DNMT3L-deficient vertebrate methylomes. Secondly, proving that LMRs is a reliable proxy of epigenetic programs, which was used to construct a chicken DNA methylation clock. Finally, verifying that inflammation led to an epigenetic age acceleration and DNA methylation may be as a marker for monitoring animal healthy status. However, there are still some questions I want to discuss with authors.

1. When you did "methylation analysis of subgenomic features", why did you choose lung, breast, sperm? I mean that immune organs are more able to reflect immune and healthy status, and pectoral muscles and leg muscles are more able to reflect production performance.

2. In table 1, n=2 in d28, I think the sample size is a bit small.

3. In Table 1, I can see age at day 3. But in methods, I did not find sample collection at day 3. Please add it.

4. The broiler breeding cycle is 42d, why only choose the time to 35d? I mean why not choose day 40 or day 42?
5. That epigenetic age acceleration means poor health has been fully confirmed by other studies?

In the end, I recommend this paper to be published with a moderate revision, if the above questions can be answered.

Reviewer #3 (Remarks to the Author):

The manuscript provides detailed insights into the chicken methylome and establishes a novel application of the DNA methylation clock as a marker for livestock health. The study is interesting, but the number of broiler samples from two different age groups was insufficient. Please explain it.

Reviewer #4 (Remarks to the Author):

Review

Article title: "A chicken DNA methylation clock for livestock health monitoring"

Brief summary

Raddatz et al. aimed to establish a chicken DNA methylation clock using low-methylated regions (LMRs) and provide evidence for its application in chicken health. They show that LMRs can be used to cluster samples by tissue type and age, and developed a chicken LMR clock with a good prediction of the chicken chronological age. The authors also generated a chicken DNA methylation dataset from different tissues and ages that will be a useful resource for further research. However, the conclusion from the experimental results are in many parts overstated and their biological interpretation, and possible applications, have not been critically evaluated and extend beyond the results. Of note is the claim that the DNA methylation clock developed can be used to measure livestock health. The authors indeed tested the DNA methylation clock for animals where inflammation was induced experimentally but the experimental methodology and data analysis needs improvement to claim that the LMR clock can be used for health monitoring. Additionally, the authors failed to connect the study of the absence of the DNMT3L cofactor in the chicken genome with the DNA methylation clock and to elucidate the development and ageing of livestock chicken. Increasing the number of tissues and developmental time points, a better validation of health measurement, and a comparison between LMR-based clock and CpG-based clock would greatly benefit the study.

Detailed Comments:

Abstract:

- "...systematically analyzed the chicken DNA methylation toolkit..." What is the meaning of the word "toolkit" in this context?
- "While overall DNA methylation patterns were similar to mammals..." "DNA methylation distribution" would be more appropriate than "DNA methylation patterns".
- "...establishes a novel application of the DNA methylation clock as a marker for livestock health." This statement is overstated.

Introduction:

- "DNA methylation clocks have already been constructed for a number of animals (e.g. mice, chimpanzees, dogs, wolves, whales), but many gaps remain with respect to the substantial differences

life spans, nutrition, breeding and other factors²⁵.” The authors could clarify what is the problem with the DNA methylation clocks of other animals or the lack of knowledge about the source of their differences. They also should mention what this work adds to the gaps in the field.

- “The broiler chicken presents a unique challenge for DNA methylation clock development, as it combines considerable economic importance with a short lifespan (42 days).” The phrase “a unique challenge” might mislead the reader into thinking the broiler chicken is not the optimal choice for this experiment.

- “...short lifespan (42 days).” Is this the real lifespan or the maximum age before the chickens are slaughtered? How does the maximum age of wild vs. livestock chicken compare?

- The authors fail to explain how the study of the methylation patterns—and the development of an epigenetic biomarker—in the chicken is representative of livestock in general. The authors should restrict the interpretation of the results to the chicken, instead of livestock.

Results:

- Fig. 1:

- Is this an analysis using multiple tissues from each species?

- The authors do not mention the presence or absence of other DNA methyltransferases/demethylases in the elephant shark or mouse.

- The figure provides weak evidence for the statement that the DNMT3L does not affect the somatic methylome in a major way. Rather, the figure only suggests that the lack of an DNMT3L homolog in the chicken, and in the elephant shark, does not prevent their methylation ratio distribution to be similar to the distribution from the mouse. The methylation dynamics in the mouse might still depend on the presence of DNMT3L in a ‘major way’. The interpretation of the figure is overstated.

- Fig. 2:

- Are the results the median of the methylation ratio in subgenomic features? Why not represent these results with a violin plot or a box plot to show the distribution? Adding error bars would help to say that figures 2a, 2b, and 2c are chicken only.

- “We also found repeats to be highly methylated (Fig. 2c), which contrasts the results from previous analyses of the same lung dataset¹⁸.” What are the differences between the analyses? And why the dataset was analysed differently?

- “Of note, we also observed moderate, but highly significant differences in the methylation levels of subgenomic features between lung, breast and sperm samples (Fig. 2a, c).” Is the high significance based on a statistical test?

- “Most notably, the sperm methylome appeared globally hypomethylated (Fig. 2d). This contrasts the sperm DNA hypermethylation in mammals³¹ (Fig. 2d)”. Could this result be due to the low number of samples (n = 1) of sperm tissue used in this study? Is this result an outlier?

- Fig 3:

- “The majority of LMRs spanned between 100 and 1000 bp and were centered around transcription factor binding sites (Fig. 3a, Fig. S1)”. The indicated figures do not show that the majority of LMRs are centered around transcription factor binding sites. The authors should add a figure that shows the number or percentage of overlaps between LMRs and transcription factor binding sites.

- Fig. 3c: the sample labels should be added to the tracks.

- Fig 4: Overall this analysis is far underdeveloped and could be expanded to include more details and properties of the chicken LMR clock. Also, the comparison to the CpG clock is barely mentioned in the text.

- “we performed additional whole-genome bisulfite sequencing of 36 samples from four tissues (breast, spleen, ileum, jejunum), with ages ranging between 3 and 35 days”. Why were those four tissues chosen?

- Fig. 4a: Which PCs are shown – please indicate in the figure.
- After the filtering and normalization step, does the PCA plot show the same age separation of the tissue samples in PC2? Please show the PCA plot after filtering and normalization.
- The figure 4b is not needed.
- “A representative example for a clock LMR is provided by the chicken *Igfbp3* locus and shows an age-dependent increase in DNA methylation (Fig. 4b)”. The described figure is the figure 4c and not 4b.
- Fig. 4e:
 - o Why the authors chose to induce inflammation using injections of CpG oligonucleotides? Would another immune trigger not be better which is unrelated to the signature of DNA methylation sites?
 - o From the methods: “Injections were administered on days 13,14,15,16,27,28,29,30 and 34 and 35”. Why the choice of only day 15, 16, and 35 for this analysis?
 - o The authors should explain the experimental setup in greater detail.
 - o Why didn’t the authors choose a biomarker to measure inflammation? A kinome peptide array together with the count of the top 10 immune related gene ontology terms might give some indication of the immune response, but should be complemented with a more reliable proxy for inflammation.
- Fig. 4f:
 - o Why in the figure 4e were shown samples from day 15 and in the figure 4f samples from day 14?
 - o Are the results statistically significant? The age prediction, using the same LMR clock, for 6 samples from the breast showed a root mean square error of 2.7 days and 3.8 days at days 14 and 28, respectively. Taking this error into consideration, the age acceleration at day 35 could be up to 3.5 days—a conservative assumption—due to error and at day 14 the age acceleration could decrease from the resulted value to 3.5 days (considering an error of, approximately, 2.7 days) for the same reason.

Reviewer #1

The purpose and results of this study is clear and easy to follow research design and its outcomes.

>> We thank the reviewer for this encouraging comment.

1. Because of small numbers of tissues and individuals, they selected only tens of methylation sites, which are relatively small considering human CpGs (<https://www.nature.com/articles/s41576-018-0004-3/>). Only jejunum was used for LMR mapping, and essentially two groups (ileum+jejunum vs. spleen+breast) for clock mapping. This study successfully found clock markers but in this current status, it would be only candidate for age prediction and health monitoring. If one choose small numbers of markers, a few markers can have large effects (LMR marker chr2:91174539-91175128, chr6:8416238-8416588, chr13:13146982-13147888). Also, some of them showed very little weights. Authors need to thoroughly investigate effects of individual markers, statistical significances and biological meanings. We can also see some high effect markers in clock CpGs.

>> This is a misunderstanding, as lung, breast and sperm were used for LMR mapping, and ileum, jejunum, spleen and breast (4 tissues, each with 3 timepoints and 3 replicates) were used for clock construction. The text and Table 1 were modified to explain this point more clearly. Also, our approach for constructing the chicken DNA methylation clock is at least comparable, if not superior to other published animal DNA methylation clocks (new Tab. S3) and thus resulted in a similar number of markers. A highly weighted example is shown in Fig. 4c, which is now clarified and further explained in the text. Also, to investigate the potential effects of individual markers, we have analyzed their enrichment in specific genomic features. The results are shown in our new Fig. 4d and show a highly interesting enrichment in promoters, which is discussed in the text.

2. Relationship between inflammation and methylation is well known, they also tried to validate LMR clock for experimentally introduced inflammation. Because their model is only focused in jejunum sample for LMR clock, they successfully showed as shown in figure 4e,f. However, it is not clear whether those putative markers also work in another tissue.

>> Because we were investigating intestinal inflammation, we used intestinal (jejunum) tissue samples for the analysis. This point has now been clarified in the text. Furthermore, we also addressed the capacity of the clock to work in other tissues in the second-to last paragraph of the discussion: "As our clock has been trained on multiple tissues, it is likely that the observed inflammation related age acceleration can also be detected in other tissues."

3. They only used penalized regression models for their study. First, selection of markers are not clearly defined. In machine learning technique, one can analyze marker effect and find optimized markers. If there are interactions (nonlinear relationships) between markers, instead of simple regression model, one can apply advanced models, i.e. deep learning methods.

>> The learning algorithm used by us is very established in the field and has been used by nearly all comparable studies, as DNA methylation dynamics is widely considered an approximately linear phenomenon. We have now clarified the approach and the selection of an optimized set of markers in the Methods section: "This approach, which computationally assigns weights to the set of CpG probes and thus selects an optimized set of markers, was established in the seminal paper by Horvath²³ and has since been applied in nearly all

studies on DNA methylation clocks.". Age prediction by other algorithms is highly unusual and was therefore not pursued.

Reviewer #2

This paper is very interesting and useful.

>> We thank the reviewer for this encouraging comment.

1. When you did "methylation analysis of subgenomic features", why did you choose lung, breast, sperm? I mean that immune organs are more able to reflect immune and healthy status, and pectoral muscles and leg muscles are more able to reflect production performance.

>> Our study starts with a basic description of the chicken methylome based on published datasets (lung, breast, sperm). In the following part, we then use tissues with production relevance (breast muscle for production, ileum and jejunum for intestinal health, spleen for overall health) to establish our multi-tissue methylation clock. This has now been clarified in the text.

2. In table 1, n=2 in d28, I think the sample size is a bit small.

>> As these data were not used for clock construction, we do not consider this a problem. Table 1 was modified to explain the usage of datasets more clearly.

3. In Table 1, I can see age at day 3. But in methods, I did not find sample collection at day 3. Please add it.

>> This has been clarified by adding a callout to Tab. 1 in the Methods section.

4. The broiler breeding cycle is 42d, why only choose the time to 35d? I mean why not choose day 40 or day 42?

>> Now clarified in the text: "The upper time point (35 days) was determined by regulatory standards for curative treatment (up to 1 week before slaughtering)".

5. That epigenetic age acceleration means poor health has been fully confirmed by other studies?

>> Yes, and we now cite a very recent review with a focus on the correlation between epigenetic age acceleration and poor health (ref. 27).

Reviewer #3

The manuscript provides detailed insights into the chicken methylome and establishes a novel application of the DNA methylation clock as a marker for livestock health. The study is interesting, but the number of broiler samples from two different age groups was insufficient. Please explain it.

>> We analyzed triplicate samples from four independent tissues and from three age groups. Table 1 was modified to explain this point more clearly. We also used whole-genome bisulfite sequencing to cover the entire methylome, while other clocks used RRBS or bisulfite PCR,

which provide much more limited coverage. Taken together, we consider this a sufficient amount of high-quality data that compares favorably to other, published animal clocks, as illustrated by our new Tab. S3.

Reviewer #4

Abstract:

1. "...systematically analyzed the chicken DNA methylation toolkit..." What is the meaning of the word "toolkit" in this context?

>> We replaced "toolkit" with "machinery".

2. "While overall DNA methylation patterns were similar to mammals..." "DNA methylation distribution" would be more appropriate than "DNA methylation patterns".

>> Corrected as suggested.

3 "...establishes a novel application of the DNA methylation clock as a marker for livestock health." This statement is overstated.

>> We replaced "establishes" with "suggests".

Introduction:

4. "DNA methylation clocks have already been constructed for a number of animals (e.g. mice, chimpanzees, dogs, wolves, whales), but many gaps remain with respect to the substantial differences life spans, nutrition, breeding and other factors²⁵." The authors could clarify what is the problem with the DNA methylation clocks of other animals or the lack of knowledge about the source of their differences. They also should mention what this work adds to the gaps in the field.

>> This has been clarified in the introduction.

5. "The broiler chicken presents a unique challenge for DNA methylation clock development, as it combines considerable economic importance with a short lifespan (42 days)." The phrase "a unique challenge" might mislead the reader into thinking the broiler chicken is not the optimal choice for this experiment.

>> The sentence was changed to: "The broiler chicken presents an important challenge for DNA methylation clock development, as it combines considerable economic importance with a short, commercially determined lifespan of only 42 days."

6. "...short lifespan (42 days)." Is this the real lifespan or the maximum age before the chickens are slaughtered? How does the maximum age of wild vs. livestock chicken compare?

>> The lifespan of broiler chickens is shortened to 42 days due to industry requirements. This has now been clarified in the text: "... a short, commercially determined lifespan of 42 days". Wild chickens have a lifespan of 5-10 years.

7. The authors fail to explain how the study of the methylation patterns—and the development of an epigenetic biomarker—in the chicken is representative of livestock in general. The authors should restrict the interpretation of the results to the chicken, instead of livestock.

>> We removed generalizations to “livestock” and restricted the interpretation of the results to chicken.

Results:

• Fig. 1:

8. Is this an analysis using multiple tissues from each species?

>> One representative tissue was used in each case. For chicken breast and sperm, this reflects the availability of published datasets. Furthermore, variations in methylation patterns between biological replicates are usually negligible on the whole-genome scale. This is also illustrated by the the 4 lung replicates in Fig. 3b, which show only a small amount of variation, even if the analysis was done the most variably methylated parts of the chicken genome (LMRs).

9. The authors do not mention the presence or absence of other DNA methyltransferases/demethylases in the elephant shark or mouse.

>> This information has been added to Fig. 1c.

10. The figure provides weak evidence for the statement that the DNMT3L does not affect the somatic methylome in a major way. Rather, the figure only suggests that the lack of an DNMT3L homolog in the chicken, and in the elephant shark, does not prevent their methylation ratio distribution to be similar to the distribution from the mouse. The methylation dynamics in the mouse might still depend on the presence of DNMT3L in a ‘major way’. The interpretation of the figure is overstated.

>> We agree and have toned down the corresponding sentence: “These findings indicate that the absence of DNMT3L does not have a major impact on the methylation ratio distribution of the somatic methylome.”

• Fig. 2:

11. Are the results the median of the methylation ratio in subgenomic features? Why not represent these results with a violin plot or a box plot to show the distribution? Adding error bars would help to say that figures 2a, 2b, and 2c are chicken only.

>> The results are now shown as violin plots, similar to other panels and figures.

12. “We also found repeats to be highly methylated (Fig. 2c), which contrasts the results from previous analyses of the same lung dataset¹⁸.” What are the differences between the analyses? And why the dataset was analysed differently?

>> For unknown reasons, the methylation status of repeats was presented ambiguously in ref. 18 and this ambiguousness has now been resolved by our analysis. To clarify this point, we have changed the wording of the sentence to: “We also found repeats to be highly methylated (Fig. 2c), which resolves ambiguous findings from previous analyses¹⁸”.

13. “Of note, we also observed moderate, but highly significant differences in the methylation levels of subgenomic features between lung, breast and sperm samples (Fig. 2a, c).” Is the high significance based on a statistical test?

>> All differences between tissues are statistically significant ($P=2.2 \times 10^{-16}$, Wilcoxon rank sum test), except the difference between lung and breast muscle at 3'-UTRs. This is now indicated in the main text and in the figure legend.

14. “Most notably, the sperm methylome appeared globally hypomethylated (Fig. 2d). This contrasts the sperm DNA hypermethylation in mammals³¹ (Fig. 2d)”. Could this result be due to the low number of samples ($n = 1$) of sperm tissue used in this study? Is this result an outlier?

>> At this point, only one chicken sperm methylome is available (ref. 19). In light of our experience with whole-genome bisulfite sequencing, “outliers” are very rare. However, with an $n=1$ sampling, this possibility cannot be ruled out. We have therefore toned down the corresponding sentences: “Most notably, the sperm methylome appeared less methylated (Fig. 2d). This contrasts the sperm DNA hypermethylation in mammals³² (Fig. 2d) and may be related to the absence of DNMT3L in chicken (see discussion).”

- Fig 3:

15. “The majority of LMRs spanned between 100 and 1000 bp and were centered around transcription factor binding sites (Fig. 3a, Fig. S1)”. The indicated figures do not show that the majority of LMRs are centered around transcription factor binding sites. The authors should add a figure that shows the number or percentage of overlaps between LMRs and transcription factor binding sites.

>> The corresponding figure has been added as Fig. S1b (top 25 transcription factor binding sites localized in LMRs).

16. Fig. 3c: the sample labels should be added to the tracks.

>> Sample labels were added to Fig. 3c.

- Fig 4

17. Overall this analysis is far underdeveloped and could be expanded to include more details and properties of the chicken LMR clock. Also, the comparison to the CpG clock is barely mentioned in the text.

>> Fig. 4 has been expanded considerably. New analyses include a PCA after normalization (Fig. 4b, see below), a comparative distribution analysis of clock markers showing a highly interesting enrichment in promoters (Fig. 4d), and a comparative validation of the LMR and CpG clocks (Fig. 4e). Also, we now show inflammation-related age acceleration for both the LMR and CpG clocks (Fig. 5c).

18. “we performed additional whole-genome bisulfite sequencing of 36 samples from four tissues (breast, spleen, ileum, jejunum), with ages ranging between 3 and 35 days”. Why were those four tissues chosen?

>> This has been clarified in the text: " The tissues were chosen because of their economic importance (breast muscle), and because of their importance for intestinal health (ileum, jejunum) and overall health (spleen)".

19. Fig. 4a: Which PCs are shown – please indicate in the figure.

>> PC1 and PC2, which is now clarified in Fig. 4a.

20. After the filtering and normalization step, does the PCA plot show the same age separation of the tissue samples in PC2? Please show the PCA plot after filtering and normalization.

>> This PCA plot is shown as our new Fig. 4b and shows a clear age separation along PC1.

21. The figure 4b is not needed.

>> We removed the panel from the figure.

22. "A representative example for a clock LMR is provided by the chicken Igfbp3 locus and shows an age-dependent increase in DNA methylation (Fig. 4b)". The described figure is the figure 4c and not 4b.

>> Corrected as suggested.

23. Fig. 4e: Why the authors chose to induce inflammation using injections of CpG oligonucleotides? Would another immune trigger not be better which is unrelated to the signature of DNA methylation sites?

>> This has now been clarified in the text: "We therefore analyzed intestinal (jejunum) samples from a trial where systemic inflammation was induced with multiple doses of a CpG oligonucleotide (Fig. 5a), which represents an established and potent protocol for the activation of cell-mediated immune responses³⁸". The mode of action of CpG oligonucleotides depends on activation of Toll-like receptors and is unrelated to DNA methylation.

24. From the methods: "Injections were administered on days 13,14,15,16,27,28,29,30 and 34 and 35". Why the choice of only day 15, 16, and 35 for this analysis?

>> As we used whole-genome bisulfite sequencing of multiple replicates for this analysis, we had to select a subset of samples. Days 15 and 16 were chosen as two independent timepoints from the early phase of the protocol, while day 35 was chosen to represent the late phase of the protocol. Also see point 25 below and our new Fig. 5a.

25. The authors should explain the experimental setup in greater detail.

>> In our new Fig. 5a, we have provided a schematic diagram of the animal trial with injection and sampling time points to clarify the experimental setup.

26. Why didn't the authors choose a biomarker to measure inflammation? A kinome peptide array together with the count of the top 10 immune related gene ontology terms might give some indication of the immune response, but should be complemented with a more reliable proxy for inflammation.

>> This has now been clarified in the text: "Kinome arrays are designed to measure the phosphorylation-based signaling of a broad range of cellular responses. Compared to individual biomarkers, like the expression of pro-inflammatory cytokines or the presence of

acute phase proteins, they allow a more comprehensive and more sensitive analysis across many species³⁹."

27. Fig. 4f: Why in the figure 4e were shown samples from day 15 and in the figure 4f samples from day 14?

>> This reflects the sample availability for independent analyses. However, all these samples are from the same age group (see our new Fig. 5a), which is now clarified in the text and figure legend.

28. Are the results statistically significant? The age prediction, using the same LMR clock, for 6 samples from the breast showed a root mean square error of 2.7 days and 3.8 days at days 14 and 28, respectively. Taking this error into consideration, the age acceleration at day 35 could be up to 3.5 days—a conservative assumption—due to error and at day 14 the age acceleration could decrease from the resulted value to 3.5 days (considering an error of, approximately, 2.7 days) for the same reason.

>> Assuming that the precision of the clock is reflected by the reproducibility of the predictions for the replicates, we used a t-test for a significant difference of the age accelerations for day 14/16 vs. age acceleration at day 35. The p-value was <0.05 in all cases, both for the LMR clock and the CpG clock. This is now indicated in Fig. 5c. Note that minor changes in the age acceleration/deceleration effects (compared to the previous manuscript version) are related to a correction in the LMR clock. Table S4 has been updated correspondingly.

REVIEWERS' COMMENTS:

Reviewer #1 (Remarks to the Author):

Authors replied all the concerns I have raised. However, there are a few things to be considered further. First, information of genes corresponding to each clock LMR and CpG (suppl. S4 and S5) need to be annotated with their functions. And, top genes related to clock LMR and CpG need to be marked accordingly not only just one example as shown in Figure 4C. Authors suggested many LMR and CpG regions for molecular clock, but only one gene function is described as for now.

Reviewer #4 (Remarks to the Author):

The revised manuscript has significantly improved and most points raised have been addressed, including a toning down of some of the stronger claims in the article. I would still strongly recommend to address three small remaining points (see below). Otherwise, despite that the manuscript does not provide major novel biological insights into the ageing process, I would recommend it for publication, based on overall sound analysis and method developed.

Remaining points

- 1) I still feel that the title is not fully reflecting the content, in particular the claim of "for livestock health monitoring" is not well enough developed to be justified in the title. There isn't enough evidence in the results to claim that this chicken methylation clock can be used for health monitoring – the abstract has been corrected in that respected. This extends to the beginning and at the end of the discussion, where the authors say "...thus establish DNA methylation age as a marker for their health status."
- 2) This manuscript has nicely shown the use of a LMR-based clock and the authors miss the opportunity to discuss more how the results from the LMR-based clock are comparable, or even better, than the CpG-based clock. This could also be highlighted it in the abstract.
- 3) The fact that a DNAm based clock can nicely predict age (and possibly health) in such an extreme animal "model" (i.e. extreme short live of the broiler chicken compared to normal chicken) is quite amazing and I would have hoped the authors would comment on this (maybe one paragraph in discussion?) - maybe they have even looked at the age prediction of their clock in samples of older normal chicken.

Reviewer #1:

Authors replied all the concerns I have raised. However, there are a few things to be considered further. First, information of genes corresponding to each clock LMR and CpG (suppl. S4 and S5) need to be annotated with their functions. And, top genes related to clock LMR and CpG need to be marked accordingly not only just one example as shown in Figure 4C. Authors suggested many LMR and CpG regions for molecular clock, but only one gene function is described as for now.

>> Obtaining a better mechanistic understanding of DNA methylation clocks is a very important research topic for the future, but it cannot be addressed in the framework of this revision. It is not trivial to annotate the genes that underpin the clocks in a meaningful way, as their numbers are too small for a pathway or gene ontology analysis and a substantial fraction of clock CpGs/LMRs resides in intergenic regions.

Reviewer #4:

1) I still feel that the title is not fully reflecting the content, in particular the claim of "for livestock health monitoring" is not well enough developed to be justified in the title. There isn't enough evidence in the results to claim that this chicken methylation clock can be used for health monitoring – the abstract has been corrected in that respect. This extends to the beginning and at the end of the discussion, where the authors say "...thus establish DNA methylation age as a marker for their health status."

>> The title was changed to: "A chicken DNA methylation clock for the prediction of broiler health".

2) This manuscript has nicely shown the use of a LMR-based clock and the authors miss the opportunity to discuss more how the results from the LMR-based clock are comparable, or even better, than the CpG-based clock. This could also be highlighted in the abstract.

>> The development of an LMR-based clock is now mentioned in the abstract.

3) The fact that a DNAm based clock can nicely predict age (and possibly health) in such an extreme animal "model" (i.e. extreme short live of the broiler chicken compared to normal chicken) is quite amazing and I would have hoped the authors would comment on this (maybe one paragraph in discussion?) - maybe they have even looked at the age prediction of their clock in samples of older normal chicken.

>> A corresponding sentence was added to the discussion: "Of note, accurate age prediction was achieved in a very short timeframe, the commercially restricted broiler chicken lifespan of 42 days".